# Model Calibration and Data Set Determination Considering the Local Micro-Structure for Short Fiber Reinforced Polymers

Andreas Primetzhofer [1,*], Gabriel Stadler [2], Gerald Pinter [3] and Florian Grün [2]

1    Polymer Competence Center Leoben GmbH, Roseggerstrasse 12, 8700 Leoben, Austria
2    Montanuniversität Leoben, Franz-Josef-Strasse 18, 8700 Leoben, Austria; Gabriel.Stadler@unileoben.ac.at (G.S.); florian.gruen@unileoben.ac.at (F.G.)
3    Montanuniversität Leoben, Otto Gloeckel-Strasse 2, 8700 Leoben, Austria; Gerald.Pinter@unileoben.ac.at
*    Correspondence: andreas.primetzhofer@pccl.at; Tel.: +43-3842-42962-53

**Abstract:** To ensure the usability of parts made of fiber-reinforced polymers, a lifetime assessment has to be made in an early stage of the development process. To describe the whole life cycle of these parts, continuous simulation chains can be used. From production to the end of the service life, all influences are mapped virtually. The later material strength is already given after the manufacturing process due to the process dependent fiber alignment. To be able to describe this fiber orientation within the lifetime assessment, this paper presents an approach for model calibration and data set determination to consider the local micro-structure. Therefore, quasi-static and cyclic tests were performed on specimens with longitudinal and transversal fiber orientation. A supplementary failure analysis provides additional information about the local micro-structure. The local fiber orientation is determined with µCT (micro computer tomography)-measurements, correlated to the extraction positions of the specimen, and implemented in a dataset. With an attached lifetime calculation on a demonstrator, a major influence of the local micro-structure on the calculation results can be shown. Therefore, it is indispensable to consider the local fiber orientation in the data set determination of short fiber reinforced polymers.

**Keywords:** lifetime calculation; fiber orientation; micro-structure; data set; fatigue

## 1. Introduction

Ever increasing requirements in terms of emission reduction, material utilization and light weight contribute to an increasing usage of short fiber reinforced thermoplastic composites instead of metals. In automotive and other industries, these materials are more and more popular for diverse load bearing applications due to their high specific strength, the cost-efficient manufacturing and recyclability. For the efficient use of reinforced composites in structural applications, under complex load conditions, it is indispensable to predict their lifetime early in the development process. Since the local material microstructure has a major influence on the mechanical behavior of inhomogeneous, anisotropic materials, such as fiber reinforced composites, all related influences must be considered in a lifetime assessment. To cover all effects during the product lifecycle, a simulation chain is established including the manufacturing process, the micro-structure simulation, and a lifetime assessment. Within this simulation chain, several models and methods are combined to predict the lifetime (depending on the material), the load case, and the ambient conditions. [1–3] These models in particular describe the influence of fiber orientation [4–6], mean stress [7,8], notch effect [9], and temperature [10–12]. In this work, special attention is paid to the local fiber orientation, as it has a major influence on the mechanical behavior. To map the influence of fiber orientation in a simulation, several models are developed and described in literature. Most of these models are based on the micro mechanical approaches, which go back to the work of Mori and Tanaka [13] and formulations for nonlinear mechanical material behavior according to Besson [14]. Over the

years, these models are extended and further developed to cover more and more influences, such as fiber interaction, fiber clustering, and process parameters [15,16]. Today's simulative description of non-linear inhomogeneous materials is mainly based on the approach of constitutive equations [17] and a homogenization of pseudo grains [18] as well as multi scale approaches [19]. A large number of other models and approaches are described in literature, which will not be further discussed here. However, most of these models focus on quasi-static material behavior and do not cover fatigue.

The lifetime estimation of short fiber reinforced polymers is mostly based on energy, damage or stress-based approaches. The former often rely on the dissipated energy during one cycle, as described in [20,21]. Herein, the dissipated energy is used as measure and compared to test results from cyclic tests. Since the dissipated energy appears as a temperature rise of the specimen, self-heating is often used to describe the fatigue behavior of a material [22–24]. Another approach to predict the fatigue life of specimen is the damage evolution. Therefore, the increase of fracture areas within the material is used in [25,26] to determine the damage. Shahabi presents in [27] a damage mechanics- based approach. Although this approach gives a deep insight of the ongoing damage processes, it is hard to transfer it to real parts. Nowadays the virtual fatigue life prediction of real parts and specimen is often based on stress-based approaches. Differently than in the so far described approaches, the lifetime is calculated by comparing a local stress and the local mechanical fatigue behavior of the material, which has to be derived from cyclic tests. The influences can then be described by different models. In [1–3,28,29] the most common stress- based simulation techniques, applicable on parts, are described. To be able to compare the actual stress to the bearable stress (stress which can be applied to reach a certain lifetime or cycles to failure), test data must be stored in some way. Primetzhofer describes in [30] a systematic approach to test and store data for the stress based lifetime estimation. In this study, the authors also describe the influence of different specimen types on the data set determination. It is shown that local fiber orientation is the most crucial and simultaneously the most difficult to describe influence on the fatigue behavior of a material.

For this study, the lifetime prediction according to the concept of local S/N-curves is used. Its application to fiber reinforced polymers is described in [1] and therefore is not explained in detail here. However, Figure 1 shows the principal procedure, which is based on commercially available software tools. While the lifetime prediction of homogenous isotropic materials such as metals follow the dashed boxes, some more steps have to be added to include the local fiber orientation.

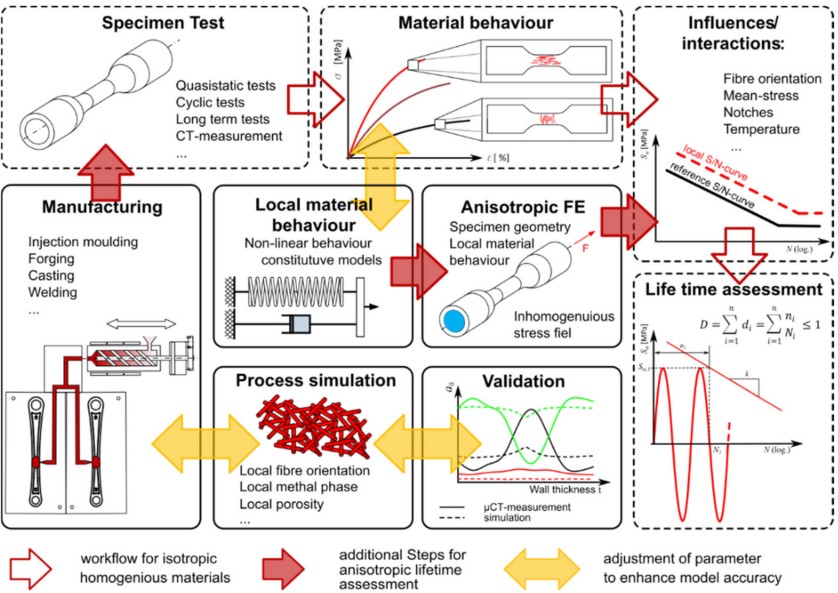

**Figure 1.** Scheme of the application of the local concept according to [1].

Special attention is paid to the interaction between the local micro-structure and the fatigue behavior and its representation in a model. Further, the influence on the data set determination of the local micro-structure is investigated in this study. To bring the local fiber orientation into play, it first has to be calculated in a process simulation and then transferred to structural simulations and the lifetime estimation. This is normally done by the symmetric fiber orientation tensor $a_{ij}$ according to Equation (1), which can be calculated directly in commercial software tools [1].

$$a = a_{ij} = \begin{bmatrix} a_{xx} & a_{xy} & a_{xz} \\ \dots & a_{yy} & a_{xz} \\ sym. & \dots & a_{zz} \end{bmatrix}. \tag{1}$$

The actual fiber orientation in respect to the loading direction is then identified by the tensor eigenvalues $\lambda$. For the further use in lifetime estimation the fatigue behavior, determined on specimens with defined fiber orientation, is related to the corresponding fiber orientation. Therefore, an exponential model according to [31] is used. Now the bearable fatigue stress $S_{FO}$ can be calculated based on the model origin (fatigue strength with no orientation) $S_0$, the local fiber orientation $\lambda$ and the model slope according Equation (2) [1].

$$S_{FO} = S_0 \cdot \exp(m \cdot \Delta\lambda). \tag{2}$$

To calculate the orientation dependent material behavior, this model is, beside others, implemented into a data set for the lifetime estimation. In this study, the effect of the local micro-structure and the corresponding fiber orientation on the fiber orientation model is investigated. Therefore, quasi-static and cyclic tests were performed on specimen extracted from injection molded plates on several positions and directions in respect to the flow direction. To verify the local fiber orientation, µCT-measurements were performed on specified plate position. Furthermore, a suggestion for model calibration and data set determination is derived.

In literature, besides the already described fiber orientation model, an approach based on the work of Tsai-Hill, the so called "Tsai-Hill-criterion", is often used to describe the mechanical behavior in respect to the loading angle of the specimen. In [30], Azzi extends this model to fatigue loads. Since this model is not used in the described simulation, this is not further discussed here.

However, by extracting specimen out of the center of plates, effects on the micro-structure in the outer regions of the plate cannot be captured. As shown in [9] for a PA66-GF 50, this effect can be significant and leads to completely different layer structures. The present work describes the application of the first described model for the extraction of specimens from the outer region of plates. On the one hand, this allows the changed micro-structure to be mapped and, on the other hand, more precise modeling for the service life calculation. Further, the impact on datasets and the associated prediction performance, especially of highly oriented domains, is described.

## 2. Materials and Methods

The investigation is performed on a short fiber reinforced polypropylene containing 50 wt% glass fibers (PP GF50). The fibers have an initial length between 3 and 5 mm, which is reduced due to processing to 0.3 to 0.5 mm. The diameter of the fibers is 10 µm. Plates with a dimension of $120 \times 80 \times 2$ mm were produced by injection molding. All sheets are manufactured with the same material and under the same process conditions. Specimens according to DIN EN ISO 527-2 [32] were milled out of these plates. An ellipsoidal transition between clamping region and designated failure area is chosen instead of a radius. This reduces the stress concentration and therefore this specimen type is especially suitable for cyclic experiments. The cross section in the failure area is $10 \times 2$ mm. In Figure 2 the specimen shape and designated failure area (parallel region in the middle of the specimen) is shown.

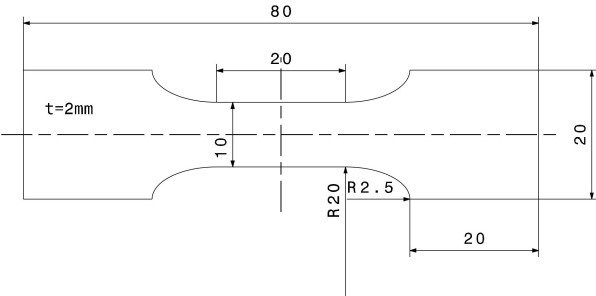

**Figure 2.** Specimen shape.

　　The specimens are milled from the plates longitudinal as well as transversal to the flow direction on different positions at the plate. A high precision CNC milling machine ensures the position. The specimens are labeled according to their position, as shown in Figure 3a. In addition, some μCT-samples are cut out from raw plates to investigate the local fiber orientation with μCT. The specimen extraction positions as well as the sample extraction points are shown in Figure 3. Each of the μCT-sample positions represents the local fiber orientation of one specimen at the expected failure position. At each position, a volume of $3 \times 2.5 \times 2$ mm is scanned with a resolution of 2 μm/voxel with GE phoenix Nanotom 180 equipped with a Molybdenum target. An acceleration voltage of 80 kV and target current of 115 μA are used for the measurement. A total of 20 measuring points were set over the thickness of the plate. The position on the plate is defined by the center of each cuboid. A similar approach is described in [33] to evaluate the fiber orientation of SMC materials. In total, five positions were measured and evaluated to map the fiber orientation distribution over the whole plate. The positions are chosen to cover the fiber orientation in the designated failure position. The positions are further labeled with T_P2 for the "top" position, M_P1 to M_P3 for the "middle" position and B_P2 for the "bottom" position. Output data were processed with the software tool Volume Graphics and further evaluated with the open- source software by FH-Wels [34]. For each measurement point, the two-order fiber orientation tensor, according to [35], can be derived. The results are presented in Section 3.3.

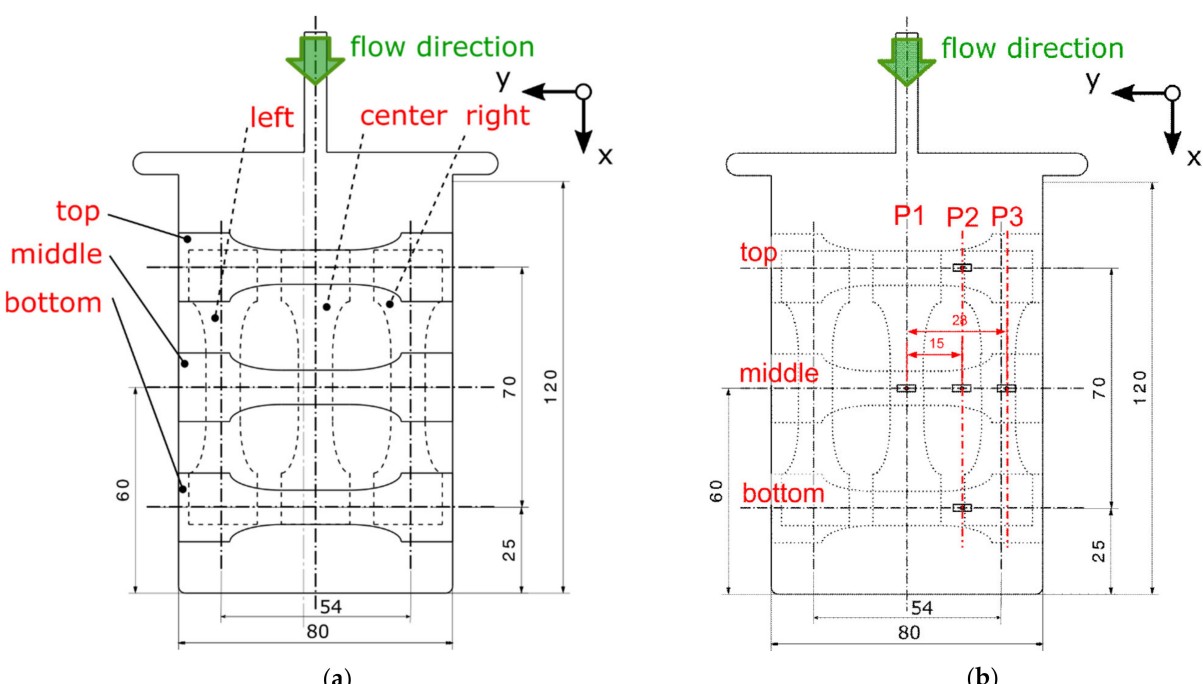

(**a**)　　　　　　　　　　　　　　　　　　　　　　　　(**b**)

**Figure 3.** Specimen extraction (**a**) and extraction position of μCT-samples (**b**).

To investigate the mechanical behavior, both quasi-static and cyclic tests were performed at each position. In previous investigations it was shown that due to the symmetrical micro-structure around the mid-plane, the results on test specimens taken from the left and right side do not differ significantly [36,37]. This is also supported by µCT-measurements on the whole plate as well as a symmetric melt flow in the cavity [38]. Since the fiber orientation is expected to be the same for the left and right position, as shown in Figure 3a, these two positions are combined in one test series. This results in five test series each for quasi-static and cyclic tests. In total, tests were performed at four extraction positions, two in longitudinal—left/right and center—as well as two in perpendicular direction—top and bottom. In addition, some specimens were tested at the middle position perpendicular to the flow direction. All tension tests were performed on an electro dynamic test rig (BOSE ElectroForce AT3550) which is equipped with a 15 kN load cell. The local strain was measured with a clip-on extensometer in the parallel part of the specimen. A loading speed of 1 mm/min was chosen according to [39] and at least three specimens were tested at each position. Test evaluation was carried out according to [39]. Cyclic tests were performed on the same machine as quasi-static test. Therefore, a sinusoidal load signal with a frequency of 3 Hz was used. To avoid hysteretic heating during the test, additional cooling was provided by a fan. In addition, the temperature was monitored by a thermocouple. As for the quasi-static tests a mechanical clamping was used. The total specimen separation or the exceeding of $N = 10^6$ cycles was defined as abort criteria. To reach cycles to failure between $10^4$ to $10^6$, three load levels were defined and at least three specimens were tested on each level to ensure statistical validation. The test evaluation was done according to [40].

## 3. Results

In the following sections, the results of quasi-static and cyclic tests are presented for each individual extraction position. Although tests were performed on all positions, at some positions the test series were reduced due to time and cost reasons. The results for all mechanical tests are normalized by the highest tensile strength derived at position "right/left".

### 3.1. Quasi-Static Tests

In Figure 4 the test results for the quasi-static test at each position are shown. Each curve represents the averaged curve of several tests. The standard deviations of the Young's modulus are between 32 and 330 MPa for all tests and between 0.2 and 0.7 MPa for the tensile strength. The test results are normalized to the highest ultimate strength, obtained for longitudinal oriented specimen on the left and right position according to Figure 3. Comparing the longitudinal specimen at the center position and left/right shows a reduction of roughly 20% in tensile strength, but just a subordinated influence on the strain at break. The modulus is also reduced by a factor of 0.85. This reduction is mainly caused by a difference in the layer structure across the plate thickness. As expected, the mechanical behavior for the transversal middle positions is nearly halved due to the transversal oriented fibers in comparison to the comparative curve. Comparing the three transversal positions, a strong influence along the flow path can be shown. With increasing distance from the injection gate, the mechanical behavior is reduced by about 30% from the uppermost to the lowest position. The significant reduced strain at break for the middle position may be due to a reduced number of tests and should be verified with further tests.

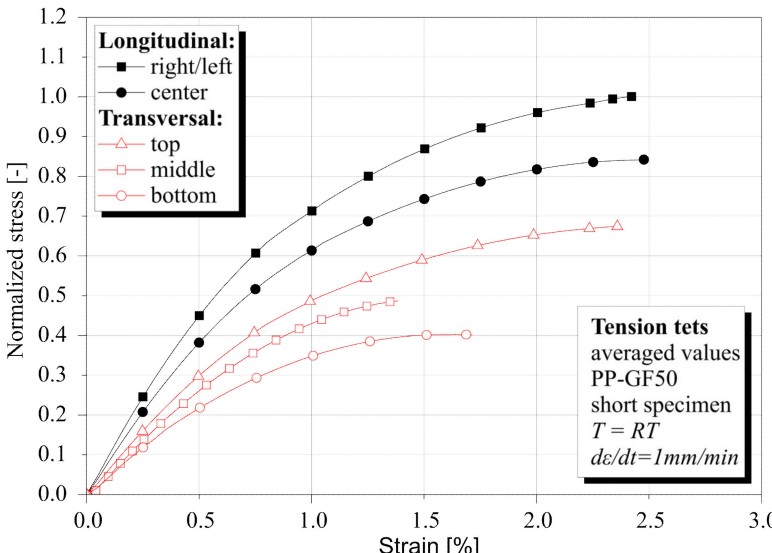

**Figure 4.** Averaged stress/strain-curves for quasi-static test results at all investigated positions.

The results show a wide range in mechanical behavior and a strong influence of the extraction position. A total reduction from the highest tensile strength to the lowest of 60% and a nearly halved Young's modulus is shown.

### 3.2. Cyclic Tests

In Figure 5, the results for the cyclic tests are shown. Herein the bearable alternating stress amplitude (R = −1) is plotted against the cycle number to failure N. The results are, normalized by the highest ultimate stress observed for "right/left"-specimen. Similar to results from quasi-static tests, a strong dependency on the extraction position can be observed. The comparison of all investigated positions shows just a light shift of the S/N-curve slopes, while the fatigue strength at $N = 10^6$ cycles drops significantly from around $S_a = 0.39$ to $S_a = 0.18$. A pronounced influence along the flow direction as well as along the plate width is observed. This is mainly caused by differences in the layer structure and the local fiber orientation. As expected, transversal specimens show a significant lower fatigue strength than longitudinal oriented.

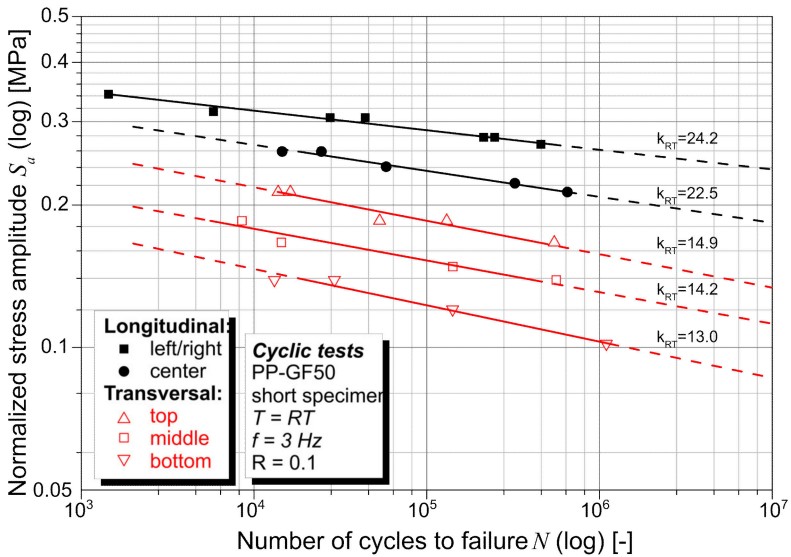

**Figure 5.** S/N-curves for quasi-static test results at all investigated positions.

Similar to the quasi-static tests, the cyclic tests also show a wide range in fatigue strength and a dependency of the extraction position.

### 3.3. Micro-Structure—µCT Measurement

In the µCT-measurement, 20 points were evaluated over the plate thickness, within the volume of the specimen. Therefore, no data point is available directly at the surface of the plate. This is justifiable insofar as the frozen layer near the surface is said to be non-aligned with small fiber content and therefore will not have a huge impact. In Figure 6, the measured values for the eigenvalue of the fiber orientation tensor, representing the fiber orientation, in flow direction are plotted against the distance from the plate surface. An automatic spline function is used to interpolate between the data points for better representation. As expected from the mechanical tests, the micro-structure shows a non-even distribution of fiber orientation and therefore a significant dependence on the extraction position. Within this non-even distribution of fiber orientation, several "layers" can be identified according to [41,42], and therefore the micro-structure is further denoted as a layered structure. Along the plate width, represented by the position "middle", a decrease in layering is observed, as shown in Figure 4. Following the curves for position 2 (T_P2—M_P2—B_P2) an increase of layering occurs. The more pronounced and more even distributed fiber orientation in flow direction for position B_P2, compared to position M_P2 and T_P2, causes a relatively low orientation in transversal direction leading to a lower mechanical properties. For the longitudinal direction, a similar observation can be drawn by comparing M_P1, representing the center specimen, and M_P3, which represents the "right/left" specimens. While a strongly layered structure is present at position M_P1, position M_P3 shows a more even distributed orientation, which in turn results in higher mechanical properties.

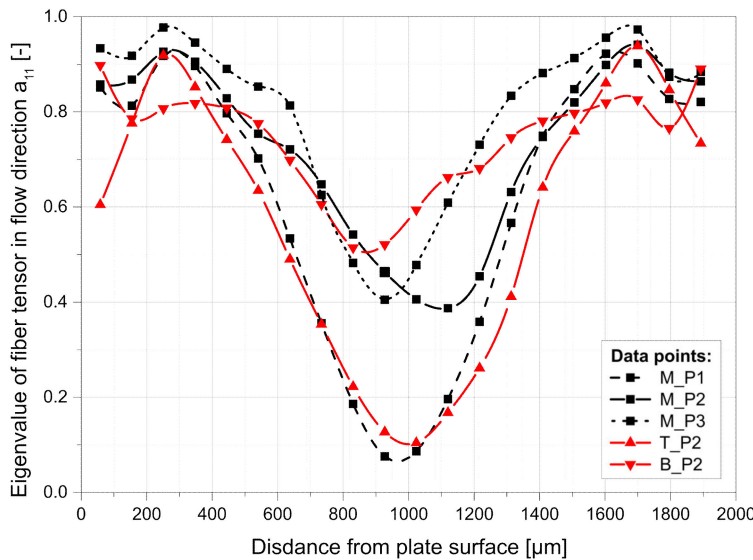

**Figure 6.** Eigenvalues of fiber orientation tensor over plate thickness at the evaluated positions.

## 4. Data Set Determination

Based on the test results, a data set for the lifetime estimation according to Figure 1 can be determined. The general approach for data set determination is described in [30]. To show the influence of the extraction position and therefore of the fiber orientation on the result on the data set, each possible combination of two positions is evaluated and compared to each other. Since the right and left position are expected to be equal and therefore tested in one test series, no separate evaluation is done for these two positions. The local fiber orientation is represented by the averaged eigenvalue of the measured orientation tensor in the longitudinal direction $\lambda_1$. For the evaluation, the tensile strength

$R_m$ and the fatigue strength $S_a$ for N = $10^6$ cycles are used. Figure 7 shows the results for both evaluations. Herein, solid lines represent the models estimated between the "left/right" position and all transversal positions. For the dotted lines, the center position is used as model point instead and compared to all transversal positions.

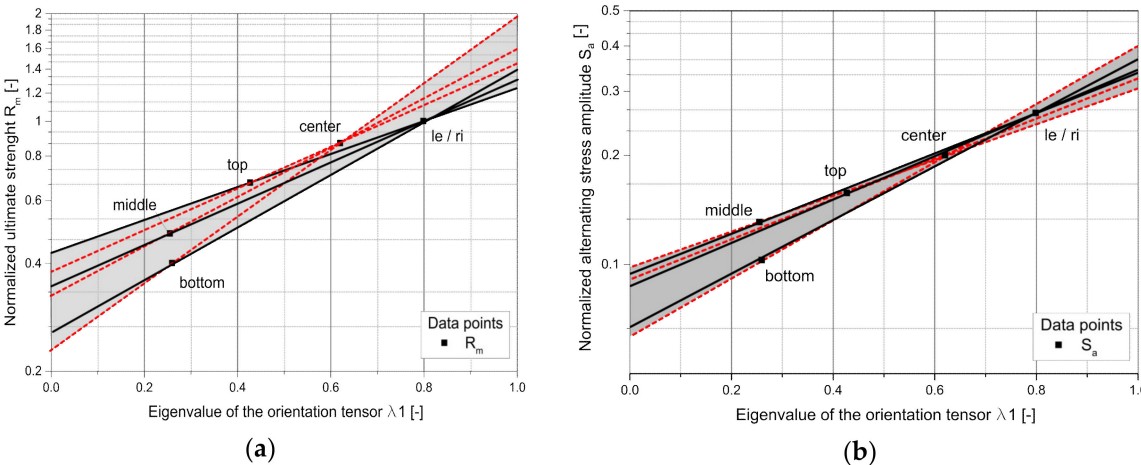

(**a**)  (**b**)

**Figure 7.** Influence of extraction position on the data set (**a**) ultimate strength (**b**) fatigue strength at N = $10^6$ cycles.

The gray shaded area thus spans the entire range of all possible combinations. Comparing the two possible model ranges (dotted lines; solid lines) for quasi-static tests (Figure 4) a significant over estimating occurs if the center position is used for modeling the fiber orientation. Therefore, too high bearable loads will be estimated in the simulation, which lead to non-conservative estimations. A much more pronounced fiber orientation in the outer regions of the plate, ensures a more narrowed spread above $\lambda_1 < 0.8$ and therefore a precise estimation in this region. In the lower range of fiber orientation ($\lambda_1 < 0.5$) a quite good correlation can be found between the two sets of models, although the dispersion of data is quite more pronounced. This is mainly caused by changes in the layer structure along the flow path. Comparing the fatigue data from cyclic tests, a more narrowed range of combination can in general be found. This can be explained by the fact that fibers act as defects. Therefore, cracks will start at these defects and grow until they merge to a critical size, leading to the specimen failure. Nevertheless, using the "center" position results in a wider range and therefore to a higher uncertainty. Depending on the preferences for a more aggressive or more conservative design one of the three curves can be chosen, without a major influence on the result at higher fiber orientations.

## 5. Discussion

The results show a strong relationship between the fiber orientation and the mechanical behavior of the investigated material and further on the data set. Beside the orientation (longitudinal/transversal), the extraction position influences the behavior. This is due to a changed micro-structure in the investigated areas. Especially the influence along the plate width, perpendicular to flow direction, is of particular interest for this study. A higher overall fiber orientation and less pronounced layered structure in the outer regions of the plate, position "left and right", is caused by a high shear flow in this region. The expansion flow in the center of the plate, "center position", will lead to a pronounced layering. This effect can be observed for all injection molded fiber reinforced materials. Lizama-Camara reports in [43] the effect for a PA6 with a fiber content of weight 50% as well as the influence of the extraction position on the mechanical behavior. Since a plate with a thickness of 4 mm is used in this study, a less pronounced layered structure is expected, compared to a thinner plate, as shown in [44]. Therefore, a smaller influence is expected than for the investigated 2 mm thick plate. In addition, the layer structure is influenced by the fiber content, as shown in [45] for a long fiber reinforced PP and PA6 and in [46]

for a short fiber reinforced polyamide with fiber contents of 30% and 60%. In general, the formation of the layer structure is determined by a complex interplay of influences. However, Lizama-Camara's study [43] shows the effect of local fiber orientation and the effect of the extraction position. This has a major influence on the material description in micro-mechanic approaches as well as macro-mechanic approaches at specimen level.

In many studies, the material calibration is done using specimens extracted in the middle of the plate (center position) under various angles (0°, 45° and 90°), as shown in [38]. Although this approach is valid, higher fiber orientations, which appear near the edges of the plate will not be covered. As very high fiber orientations are expected in components, especially in areas with very thin walls, this can lead to deviations in the design. In addition, different fiber contents and other influences may change the layer structure in the center of the plate so that a more or less pronounced layer structure arises in this region. Although this also affects the outer region of the plate, the effect will be less pronounced due to less layering in this region. Using the angle between the main fiber orientation and the loading direction as a measure in a model hinders the transferability to simulative methods, since the fiber orientation is just available as a second order tensor in process-simulation [47,48]. This applies for complex 3D structures in particular, because neither the fiber orientation can be described with one angle nor the load can be described in respect to the fiber orientation.

Therefore, the local fiber orientation tensor has to be considered in the lifetime assessment of short fiber reinforced materials, as described in Section 1, according to Equation (2). Herein the fiber orientation is described by the eigenvalue of the second order tensor. To cover a wide range of fiber orientations, the authors recommend the extraction of specimens in the left and right position for the longitudinal and the middle position for the transversal direction. As shown in this work, this will lead to a good balance between aggressive and conservative estimations for the lifetime assessment. Since the use of the center position as a model point leads to an overprediction for higher fiber orientations, this is not recommended.

However, this study, as many others [4,49–51], also shows a major influence of the layer structure, the extraction direction and position on the mechanical behavior. Although the presented approach uses specimens with higher fiber orientations extracted from the outer regions of the plate, transversal specimens stretch over the center position as in all other studies. Due to the changing layer structure over the plate width, it is hard to define the right fiber orientation for model calibration. Although the given approach leads to good results in the lifetime estimation of fiber reinforced materials, the influences of the local fiber orientation and in particular the layer structure have to be investigated in more detail in the future.

**Author Contributions:** Conceptualization: A.P.; methodology: A.P.; validation: A.P., G.S.; formal analysis: A.P., G.S.; investigation: A.P., G.S.; resources: G.S.; data curation: A.P.; writing—original draft preparation: A.P.; writing—review and editing: A.P., G.S., G.P.; visualization: A.P., G.S.; supervision: G.P.; F.G.; project administration: A.P.; funding acquisition: A.P., G.P. All authors have read and agreed to the published version of the manuscript.

**Funding:** This research was funded by FFG grant number 854178. The research work of this paper was performed by the Polymer Competence Center Leoben GmbH (PCCL) in collaboration with the Chair of Mechanical Engineering and the Chair of Science and Testing of Polymers at the Montanuniversiaet, within the framework of the COMET-program of the Austrian Ministry of Traffic, Innovation and Technology, with contribution by BOREALIS AG, Engineering Center Steyr GmbH (MAGNA Powertrain ECS), EVONIK Industries AG, Volkswagen AG, Schaeffler Technologies AG and Co KG and Mann and Hummel AG. The PCCL is founded by the Austrian Government and the State Governments of Styria, Lower and Upper Austria.

**Institutional Review Board Statement:** Not applicable.

**Informed Consent Statement:** Not applicable.

**Data Availability Statement:** The data presented in this study are available on request from the corresponding author.

**Conflicts of Interest:** The authors declare no conflict of interest. The funders had no role in the design of the study; in the collection, analyses, or interpretation of data; in the writing of the manuscript, and in the decision to publish the results.

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
