# Peer review of "Model Calibration and Data Set Determination Considering the Local Micro-Structure for Short Fiber Reinforced Polymers"

_jcs, doi:10.3390/jcs5020040_

Round 1

Reviewer 1 Report

This manuscript dealt with model calibration and data set determination considering the local micro structure for short fibre reinforced polymers. The idea is interesting and the results are useful. However, there lacks details in many parts of this manuscript. For example, for the material used, what is the fibre diameter and target length? there is no mention what method the authors used to describe the fiber orientation tensor, i.e. how were the results of eigenvalues in Figure 6 determined? Some details of the method should be described here. It is well known that second order fibre orientation tensor introduced by Advani S, Tucker III C (J Rheol 1987;31:751-784) is widely used in the community but was not mentioned at all here, can the authors comment? Recently, this method has been used for fiber composite materials, see Sabiston T, et al, Composite, Part A, 114, 278-294, 2018 and in particular fatigue of CFRP in Sabiston T, et al, Composite Structure, 2021.  

Author Response

Dear reviewer,

Thank you for the comments, which were very helpful to improve the quality of the present paper. We add a comment to each reviewers-comment (Remark. 1, ..) followed by an answer and the changed or added text itself.

Remark 1: For example, for the material used, what is the fibre diameter and target length?

Answer: A note is added about the fiber diameter and the target length.

Added: The fibers have an initial length between 3 and 5mm which is reduced due to processing to 0.3 to 0.5 mm. The diameter of the fibers is 10µm.

Remark 2: there is no mention what method the authors used to describe the fiber orientation tensor, i.e. how were the results of eigenvalues in Figure 6 determined? Some details of the method should be described here. It is well known that second order fibre orientation tensor introduced by Advani S, Tucker III C (J Rheol 1987;31:751-784) is widely used in the community but was not mentioned at all here, can the authors comment? Recently, this method has been used for fiber composite materials, see Sabiston T, et al, Composite, Part A, 114, 278-294, 2018 and in particular fatigue of CFRP in Sabiston T, et al, Composite Structure, 2021.  

Answer: The method (µCT-Measurements) is already described in section 2 (Materials and Methods) of the paper. However, some more details are added to clarify the method.

Added: The specimen extraction positions as well as the sample extraction point are shown in Figure 3. Each of the µCT-sample positions represents the local fiber orientation of one particular specimen at the expected failure position. At each position, a volume of 3x2,5 mm is scanned with a resolution of 2µm/voxel with GE phoenix Nanotom 180 equipped with a Molybdenum target. An acceleration voltage of 80kV and target current of 115µA are used for measurement. In total 20 measuring points were set over the thickness of the plate. The position on the plate is defined by the center of each cuboid. A similar approach is described in [1] to evaluate the fiber orientation of SMC materials. In total five positions were measured and evaluated to map the fiber orientation distribution over the whole plate. The positions are chosen to cover the fiber orientation in the designated failure position. The positions are further labeled with T_P2 for the “top” position, M_P1 to M_P3 for the “middle” position and B_P2 for the “bottom” position. The output data form the measurement were processed with Volume Graphics software and further evaluated with the open source software by FH-Wels [2]. For each measurement point the 2-order fiber orientation tensor according to [3] can be derived.  The results are presented in section 3.2.

Reviewer 2 Report

The authors presented interesting work on the influence of fiber orientation on short fiber composites' fatigue life. Although the approach is interesting, a few things are missing in the manuscript.

Firstly, in the manuscript, there are a few missing commas, and some wording is unclear. A simple grammar check and a reread will solve these issues. Furthermore, referencing and cross-referencing is not consistent. At times, it is with et al., and sometimes, it's just the first author. Unification will be easier to read.

Line 30-31: Short fiber-reinforced composites for structural applications? This is not sure. 

Line 45: Mori and Tanaka
Line 46: Besson et al.
Line 48: What are those influences?
Lines 49-52: The intent is unclear. 
Line 68: What does bearable stress mean?

Equation 1: it should be a= and not aij the parenthesis are missing

Line 130: The volume is not clear. Just two dimensions are mentioned, and how were these samples extracted? Were the scans repeated with other samples?

Line 138: The authors expected the same behavior on the right and left sides of the specimen. But no validation was presented in the manuscript. Was it just an assumption, or was it reasoned based on the preliminary tests?

Line 153: What are the three load levels?

In figure 4: Was only one test conducted for each case? If so, the conclusion drawn can be speculative. Can authors provide the values of standard deviations for each case?

Furthermore, if there is a difference of 20% of tensile strength between right/left and center, it is just because of the fiber orientation or the fiber volume fraction in the specimen.

What exactly is a layered structure? These short-fiber composites cannot be considered as laminates.

Can authors also provide the local fiber volume fractions for each case?

In Figure 5, do the values represent a single test or the mean value? Then the repeatability of the tests can be questioned. 

Line 191: What was the segmentation tool and technique? Why only 20 points were analyzed instead of the total thickness of the specimen?

The values presented in Figure 6 represent the mean value or single value? How many fibers were considered to report each eigenvalue in Figure 6? Is it possible for authors to provide a histogram? It would be helpful to understand the population of fibers for each eigenvalue.

Figure 7: Can authors bring in some values from literature to further support their envelope?

Line 256: reference is missing

Line 273: Is there an interest for authors in suggesting a mathematical model based on the orientation tensor and fatigue life?

Line 282: PMP16?

Line 286: Layered structure?

Author Response

Dear reviewer,

Thank you for the comments, which were very helpful to improve the quality of the present paper. We add a comment to each reviewers-comment (Remark. 1, ..) followed by an answer and the changed or added text itself.

Remark 1: Firstly, in the manuscript, there are a few missing commas, and some wording is unclear. A simple grammar check and a reread will solve these issues. Furthermore, referencing and cross-referencing is not consistent. At times, it is with et al., and sometimes, it's just the first author. Unification will be easier to read.

Answer: The text has been updated with regard to commas and wording. The citations are unified.

Added: -

Remark 2: Short fiber-reinforced composites for structural applications? This is not sure. .

Answer: We agree that it is not sure that short fiber reinforced composites are used for structural applications. However, they are and will be used for load bearing applications instead just being covers or other non-loaded parts.

Changes: In automotive and other industries these materials are more and more popular for diverse load bearing applications due to their high specific strength, the cost-efficient manufacturing and recyclability

Remark 3: Mori and Tanaka

Answer:

Changes: Mori Tanaka à Mori and Tanaka

Remark 4: Besson et al

Answer:

Changes: Besson et al à Besson

Remark 4: What are those influences

Answer: some influences and citations are added

Changes: Over the years, these models are extended and further developed to cover more and more influences like fiber interaction, fiber clustering and processparameter [1, 2].

Remark 5: The intent is unclear

Answer: Short remark on which methods are currently used to describe the material behavior in a simulation.

Changes: No changes. 

Remark 6: What does bearable stress mean?

Answer: bearable means the stress which can be applied to reach a certain life time or cycles to failure

Changes: A short description is added

…. Bearable stress (stress which can be applied to reach a certain life time or cycles to failure) ….

Remark 7: Equation 1: it should be a= and not aij the parenthesis are missing

Answer: Equation is supplemented with parenthesis and a is add to the equation; the notation aij is already used for other publications and therefore also used here to be consistent. However, a remark to the related publication is added

Changes: a is added in the equation; parenthesis are added; reference is added in the text right before equation 1

Remark 8: The volume is not clear. Just two dimensions are mentioned, and how were these samples extracted? Were the scans repeated with other samples?

Answer: The third dimesion is added to the text. Further the text is slightly changed to clarify the extraction method and which samples are used for the µCT-specimens

Changes: The specimens are milled from the plates longitudinal as well as transversal to the flow direction on different positions at the plate. The specimens are labeled according to their position, as shown in Figure 3 (a). In addition, some µCT-samples are cut out from virgin plates to investigate the local fiber orientation with µCT. The …..

position. At each position, a volume of 3x2,5x2 mm is scanned with a resolution of 2µm/voxel with GE phoenix Nanotom 180 equipped with a Molybdenum target

Remark 9: The authors expected the same behavior on the right and left sides of the specimen. But no validation was presented in the manuscript. Was it just an assumption, or was it reasoned based on the preliminary tests?

Answer: In previous investigations on a similar material as well as on other materials it could be proven that the left and right position are comparable. Some explanation is added in the text.

Changes: In previous investigations it was shown that due to the symmetrical microstructure around the mid-plane, the results on test specimens taken from the left and right side do not differ significantly [3, 4]. This is also supported by µCT-measurements on the whole plate as well as a symmetric melt flow in the cavity [5].

Remark 10: What are the three load levels?

Answer: the three load levels are different for each removal position and are selected to achieve cycles to break between 104 and 106. This is already described in the publication.

“To reach cycles to failure between 104 to 106 three load levels were defined and at least three specimens were tested on each level to ensure statistical validation. “

Changes: No changes

Remark 11.1: In figure 4: Was only one test conducted for each case? If so, the conclusion drawn can be speculative. Can authors provide the values of standard deviations for each case?

Answer: Figure 4 show the averaged curve for several tests, as mentioned in the text. The standard deviation range for Young’s Modulus and ultimate tensile strength are added in the text.

Changes: …. several tests. The standard deviations of the Young's modulus are between 32 and 330 MPa for all tests and between 0.2 and 0.7 MPa for the tensile strength. The test ….

Remark 11.2: Furthermore, if there is a difference of 20% of tensile strength between right/left and center, it is just because of the fiber orientation or the fiber volume fraction in the specimen.

Answer: XXX

Remark 11.3: What exactly is a layered structure? These short-fiber composites cannot be considered as laminates.

Answer: Due to stresses the fibers are aligned within the melt during the injection process. This results in the formation of zones with different fiber orientation, which can then be referred to as different layers. An explanation has been added to the text.

Changes: … position. Within this non-even distribution of fiber orientation several “layers” can be identified according to [6, 7], and therefore the microstructure is further denoted as layered structure ….

Remark 11.3: Can authors also provide the local fiber volume fractions for each case?

Answer: The present approach bases only on the fiber orientation, therefore the fiber volume fraction is not described here in detail. See also Remark 14.

Changes: No changes

Remark 12: In Figure 5, do the values represent a single test or the mean value? Then the repeatability of the tests can be questioned. 

Answer: Each value represent a single values (or sometimes two or more – cannot be distinguished due to almost similar cycles to failure). The tests are conducted according ASTM739 which is a standard approach for S/N-curve determination.

Changes: No changes

Remark 13: Line 191: What was the segmentation tool and technique? Why only 20 points were analyzed instead of the total thickness of the specimen?

Answer: The number of 20 measuring points results from the selected resolution, which has proven itself for the fiber orientation measurement. This also results in an even distribution over the entire slab thickness, as shown in Figure 6. A small remark is added in the Text.

Changes: … Therefore, no data point is available directly at the surface of the plate. This ….

Remark 14: The values presented in Figure 6 represent the mean value or single value? How many fibers were considered to report each eigenvalue in Figure 6? Is it possible for authors to provide a histogram? It would be helpful to understand the population of fibers for each eigenvalue.

Answer: During the measuring process, the volume to be measured is divided into smaller sections (in this case 20). The values shown represent the average orientation in this volume and therefore include a large number of fibers. The The precise evaluation of the fiber content / fiber volume fraction was refrained from here, since the approach is based on the fiber orientation only and therefore a detailed description of these influences is beyond the focus of this work. We agree with you that the fiber content / fiber volume fraction has an influence on the mechanical behavior. We are currently working on a study on this and will publish it after completion.

Changes: No changes

Remark 15: Figure 7: Can authors bring in some values from literature to further support their envelope?

Answer: The results shown in Figure 7 are intended to demonstrate and create awareness of the principle influence of local fiber orientation. Additional points for other materials would reduce, in the authors' view, the clarity of the diagram and thus the significance. Similar studies have already been carried out for other materials or are currently underway and will be published later.   

Changes: No changes

Remark 16: Line 256: reference is missing

Answer: Reference is added

Changes: However, Lizama-Camara’s study [45] shows the effect of local fiber orientation and the effect of the extraction position. This has a major influence

Remark 17: Line 273: Is there an interest for authors in suggesting a mathematical model based on the orientation tensor and fatigue life?

Answer: The Fiber orientation tensor is already considered in the life time assessment of short fiber reinforced polymers according to equation 2 as described in this work.

Changes: No changes

Remark 18: Line 282: PMP16?

Answer: Formation issue with the automated citation - changed.

Changes: … study, as many others [51, 4, 52], also shows a major …

Remark 19: Line 286: Layered structure?

Answer: A description of what is meant by layered structure is added in the text – see Remark 11.3

Changes: no changes

Round 2

Reviewer 1 Report

This revision addresses most of my comments. There are two left 1. There is a recent publication on fatigue of short fiber CFRP considering local microstructure and orientations by Sabiston et al that the authors should cite: https://doi.org/10.1016/j.compstruct.2020.112898 2. The authors should label a coordinate system somewhere in Fig. 1 and confirm that they are all consistent in their various descriptions in experiments and modeling. 3. How did the authors maintain their register in XCT and align it with the rest experiments to ensure there is no overlap or overlook in their orientation data. This is not clearly described either.

Author Response

Remark 1: There is a recent publication on fatigue of short fiber CFRP considering local microstructure and orientations by Sabiston et al that the authors should cite: https://doi.org/10.1016/j.compstruct.2020.112898

Answer: A citation is added in the discussion

Changes: … many others [51, 4, 52, 53], also shows ….

Remark 2: The authors should label a coordinate system somewhere in Fig. 1 and confirm that they are all consistent in their various descriptions in experiments and modeling

Answer: Figure 1 describe the process itself, therefore no coordinate system is added. Instead, a coordinate system is added to figure 3 where the position of all specimen is described. In addition, a remark about the consistency is added in the text.

Changes: ….. To ensure consistency in further data determination and data evaluation all results are referred to the coordinate system, showed in Figure 3. ….

Remark 3: How did the authors maintain their register in XCT and align it with the rest experiments to ensure there is no overlap or overlook in their orientation data. This is not clearly described either.

Answer: The µCT specimen were cut from defined positions on separate virgin plates as described in the paper. These positions represents the fiber orientation in the designated failure area on the specimen. All other specimen were extracted at defined positions by milling. A high precision CNC milling machine ensures the position. All sheets are manufactured with the same material and under the same process conditions. this ensures a consistent fiber distribution in all sheets. A remark about this is added in the text. The uniform process and accurate sampling ensures the µCT measurements register and the alignment to all other experiments.

Changes: … mension of 120x80x2 mm were produced by injection molding. All sheets are manufactured with the same material and under the same process conditions. This ensures a consistent fiber distribution in all sheets. Specimen according…

… longitudinal as well as transversal to the flow direction on different positions at the plate. A high precision CNC milling machine ensures the position. The specimens are labeled according to their position, as …

Reviewer 2 Report

-

Author Response

Dear reviewer,

thank you for review.

Best Andreas Primetzhofer
